# An E280K Missense Variant in *KCND3*/Kv4.3—Case Report and Functional Characterization

**DOI:** 10.3390/ijms241310924

**Published:** 2023-06-30

**Authors:** Richard Ågren, Niels Geerdink, Han G. Brunner, Martin Paucar, Erik-Jan Kamsteeg, Kristoffer Sahlholm

**Affiliations:** 1Department of Physiology and Pharmacology, Karolinska Institutet, 171 77 Stockholm, Sweden; richard.agren@ki.se; 2Department of Pediatrics, Rijnstate Hospital, 6815 AD Arnhem, The Netherlands; ngeerdink@rijnstate.nl; 3Department of Human Genetics, Donders Centre for Brain, Cognition and Behavior, Radboud University Medical Center, 6525 GA Nijmegen, The Netherlands; han.brunner@radboudumc.nl; 4Department of Clinical Genetics, MUMC Maastricht, GROW School for Oncology and Developmental Biology, MHENS School for Mental Health and Neuroscience, Maastricht University Medical Center, 6229 HX Maastricht, The Netherlands; 5Department of Neurology, Karolinska University Hospital, 141 86 Stockholm, Sweden; martin.paucar-arce@regionstockholm.se; 6Department of Clinical Neuroscience, Karolinska Institutet, 171 77 Stockholm, Sweden; 7Department of Human Genetics, Radboud UMC, 6525 GA Nijmegen, The Netherlands; erik-jan.kamsteeg@radboudumc.nl; 8Department of Integrative Medical Biology, Wallenberg Centre for Molecular Medicine, Umeå University, 901 87 Umeå, Sweden

**Keywords:** spinocerebellar ataxia type 19/22, episodic ataxia, voltage-gated potassium channel D3, electrophysiology, mild developmental delay, rare variants, voltage sensor

## Abstract

A five-year-old girl presented with headache attacks, clumsiness, and a history of transient gait disturbances. She and her father, mother, twin sister, and brother underwent neurological evaluation, neuroimaging, and exome sequencing covering 357 genes associated with movement disorders. Sequencing revealed the new variant *KCND3* c.838G>A, p.E280K in the father and sisters, but not in the mother and brother. *KCND3* encodes voltage-gated potassium channel D3 (Kv4.3) and mutations have been associated with spinocerebellar ataxia type 19/22 (SCA19/22) and cardiac arrhythmias. SCA19/22 is characterized by ataxia, Parkinsonism, peripheral neuropathy, and sometimes, intellectual disability. Neuroimaging, EEG, and ECG were unremarkable. Mild developmental delay with impaired fluid reasoning was observed in both sisters, but not in the brother. None of the family members demonstrated ataxia or parkinsonism. In *Xenopus* oocyte electrophysiology experiments, E280K was associated with a rightward shift in the Kv4.3 voltage-activation relationship of 11 mV for WT/E280K and +17 mV for E280K/E280K relative to WT/WT. Steady-state inactivation was similarly right-shifted. Maximal peak current amplitudes were similar for WT/WT, WT/E280K, and E280K/E280K. Our data indicate that Kv4.3 E280K affects channel activation and inactivation and is associated with developmental delay. However, E280K appears to be relatively benign considering it does not result in overt ataxia.

## 1. Introduction

*KCND3* encodes the voltage-gated potassium D3 (Kv4.3) channel, which is highly expressed in the cerebellum and heart [1,2]. The functional channel is formed by four Kv4.3 subunits, each consisting of six transmembrane segments (S1–S6). Spinocerebellar ataxia 19 (SCA19), allelic with spinocerebellar ataxia 22 (SCA22), is associated with *KCND3* mutations, and so far, over 20 mutations have been reported [3,4,5,6,7,8,9,10,11,12,13,14]. The SCA19/22 phenotype is variable and includes, besides ataxia and motor features, varying degrees of intellectual disability and epilepsy. Neuroimaging in SCA19/22 demonstrates cerebellar atrophy, whereas positron emission tomography studies have demonstrated various degrees of cerebral and cerebellar hypometabolism [9,11]. In addition, *KCND3*/Kv4.3 missense mutations have been reported in association with cardiac arrhythmias, including Brugada syndrome [15,16,17].

Functional analyses of SCA19/22-associated *KCND3* mutations indicate either channel loss-of-function or reduced cell membrane trafficking, which for some mutations was partly counteracted by the co-expression of the potassium-channel-interacting protein 2 (KChIP2) [18]. In addition to protein-truncating mutations, reported missense mutations are frequently located in S5-6, which flank the ion-conducting pore loop. However, a deletion at F227 [3] and a duplication of three residues in the voltage sensor, S4, have also been reported [5]. To the best of our knowledge, no missense mutations in the S3–S4 loop have been reported for *KCND3*/Kv4.3. 

Here, we characterize a family in which one daughter is affected by episodic gait disturbances and mild developmental delay. Analysis with exome sequencing revealed a novel *KCND3* c.838G>A variant in the index patient, her monozygotic twin sister, and their father. This nucleotide substitution results in E280K missense mutation of the Kv4.3 protein. We also describe the functional properties of this variant in *Xenopus* oocytes using two-electrode voltage clamp.

## 2. Results

### 2.1. Clinical Findings 

The index patient was born prematurely at a gestational age of 34 weeks. At the age of 3 years, she had episodes with headache attacks and vomiting, which raised the suspicion of migraine. In addition, the parents reported short episodes of loss of consciousness with deviation of the eyes sometimes accompanied by drop attacks or gait disturbances. The index patient displayed motion sickness and avoided rocking and turning movements. Such angular accelerations appeared to provoke dizziness and gait disturbances, suggesting a possible episodic ataxia or another paroxysmal movement disorder. These episodes gradually resolved spontaneously after one year and a half, but a mild motor delay remained (second percentile score on Movement Assessment Battery for Children at the age of 5 years; see Appendix A). The patient displayed recurring difficulties with static balance, but her gait was normal, and there were no signs of dysmetria or nystagmus upon examination. Neuroimaging was unremarkable. Repeated EEGs were normal, as were the cardiological findings.

The identical twin sister of the index patient carried the E280K variant but did not have headache attacks or gait disturbances. During clinical evaluation, no ataxia was observed. Like her sister, she had a mild motor delay (first percentile score on Movement Assessment Battery for Children at the age of 5 years; Appendix A). The father of the twin sisters, who carried the E280K mutation (Figure 1A), reported that he had transient episodes of gait disturbances with severe vertigo and vomiting since the age of 39 years. These episodes lasted a few days. Evaluation by a neurologist did not result in a clear diagnosis and neuroimaging was unremarkable. The mother and brother of the index case did not carry the E280K variant. An extensive multidisciplinary evaluation of the twin sisters and brother was performed when the sisters were aged five. These examinations demonstrated a mild global developmental delay in the twin sisters compared to the brother, which was most pronounced for fluid reasoning (Figure 1B). 

### 2.2. Genetic Analyses

The index patient, her sister, and father were found to be heterozygous for a *KCND3* c.838G>A, p.E280K variant, which has not been reported earlier. In the translated protein, the resulting mutant residue is in the second extracellular loop between S3 and S4, which is located on the extracellular face of the channel complex and adjacent to the voltage sensor, S4 (Figure 2A,B). 

### 2.3. Electrophysiology Investigations of KCND3 E280K 

*Xenopus* oocytes were injected with a constant amount of cRNA encoding WT Kv4.3 and/or the E280K variant to mimic the WT, heterozygous (WT/E280K), and homozygous conditions (E280K). cRNA encoding the Kv channel-interacting protein 2 (KChIP2) was coinjected with Kv4.3 cRNA in all conditions, resulting in depolarization-evoked outward currents with swift inactivation (Figure 3A), consistent with previous reports [18]. The maximal peak current amplitudes at +60 mV were similar between oocytes expressing WT Kv4.3, WT/E280K, and E280K. However, current responses were reduced (approximately −30 mV) for WT/E280K and E280K compared to WT (Figure 3A,B). The time to peak current increased for E280K (49.9 ± 1.4 ms) at −30 mV compared to WT (34.0 ± 0.6 ms; *p* = 0.0036, two-way ANOVA; Figure 3C). Also, deactivation was slowed for E280K compared to WT at 0–30 mV (Figure 3D). The corresponding normalized conductance–voltage relationships were increasingly right-shifted for WT/E280K and E280K (Figure 3E). Fitting of Boltzmann curves to the conductance–voltage relationships revealed significant right-shifting in the half-activation voltages from −27.2 ± 0.4 mV (WT) to −16.3 ± 0.3 mV (WT/E280K; *p* < 0.0001, ANOVA vs. WT) and −10.5 ± 0.4 mV (E280K; *p* < 0.0001, ANOVA vs. WT). 

Steady-state inactivation was investigated by voltage-stepping to +50 mV from membrane potentials between −100 mV and +60 mV. Sigmoidal curve fitting to the steady-state inactivation–voltage relationship revealed that the half steady-state inactivation potential of Kv4.3 was −64.3 ± 0.2 mV for WT, −55.6 ± 0.2 mV for WT/E280K (+8.7 mV compared to WT, *p* < 0.0001, ANOVA), and −48.6 ± 0.2 mV for E280K (+15.7 mV compared to WT, *p* < 0.0001, ANOVA (Figure 4A–C)). 

## 3. Discussion

The index case (a five-year-old girl) demonstrated transient episodes with gait disturbances that prompted genetic testing for episodic ataxia. The heterozygous Kv4.3 (*KCND3*) E280K variant was found in the index patient, her monozygotic sibling, and their father. Although the (family) history suggested an episodic ataxia, no ataxia was observed on neurological examination or on other clinical observations. However, cognitive tests indicated reduced fluid reasoning in the twin sisters carrying the mutation. 

The E280K mutation is located on the extracellular face of the Kv4.3 channel, in the S3–S4 linker. The adjacent voltage sensing S4 segment contains positively charged residues and is displaced in the extracellular direction upon membrane depolarization. S4 movement evokes a conformational change in the protein that leads to channel opening. Electric charges located in the extracellular loops have been found to influence the behavior of the S4 segment, with an increased positive charge on the extracellular surface of the channel causing repulsion of the positively charged S4 segment and a consequent rightward shift in the voltage–conductance curve [19,20,21], i.e., stronger depolarization is needed to open the channel. Of particular relevance here is that deleting negatively charged residues in the N-terminal S3–S4 linker of the Drosophila eag (ether-à-go-go) channel was reported to shift the activation to more positive potentials [22,23]. 

Similarly, here we found that the E280K substitution induced a rightward shift in the activation midpoint potential of 17 mV relative to WT. Earlier studies furthermore indicate that voltage sensor movement is coupled not only to activation, but also to inactivation of Kv4 channels [24,25]. This is consistent with the present observation that E280K mutation also shifted the inactivation midpoint potential of Kv4.3. However, the inactivation kinetics that we observe in our oocyte system are slower than those typically observed in mammalian cells, which limits the interpretation of this aspect of channel function. Thus, future studies should be performed in mammalian cells to further address the role of E280K on Kv4.3 fast inactivation. While the large currents recorded in our study raise the potential for series resistance errors, any such perturbations should affect WT and mutant channels to similar extents (due to their similar current amplitudes) and would not qualitatively alter the results of this study.

The location of the *KCND3* c.838G>A, p.E280K variant in the S3–S4 linker is unique, and most of the previously described missense mutations are located in, or adjacent to, the ion selectivity filter between S5–S6. A recent report described a nine-nucleotide duplication in *KCND3*, leading to three extra residues, including a positively charged arginine in the voltage-sensing S4 segment and a pronounced (~+65 mV) rightward shift of the activation curve [5]. Interestingly, in contrast to the present findings, that KCDN3 variant was associated with ataxia and epilepsy, presumably linked to the greater functional perturbation of the Kv4.3 protein. 

The similar maximal peak current amplitudes of Kv4.3 WT and E280K (Figure 3) suggest that this channel does not affect K+ ion passage and retains WT levels of membrane expression. This is contrary to several Kv4.3 variants associated with SCA19/22, which showed markedly reduced currents or were unable to form functional channels, exhibiting dominant-negative effects when co-expressed with WT Kv4.3 [4,11,12]. Thus, the rightward shift in the activation curve observed with E280K variant described here may not sufficiently perturb cell excitability so as to cause ataxia, epilepsy, or cardiac arrhythmia, although it may contribute to intellectual impairment. However, a complex relation between phenotype and genotype with reduced penetrance has previously been described for a R419H *KCND3* variant, based on data from four individuals in the gnomAD database [13]. Additionally, age-related progression of symptoms has been reported for SCA19/22 [14]. Thus, one cannot rule out the possibility that the E280K mutation could cause more severe symptoms in other contexts or individuals than those examined here. An important limitation of this study is that we cannot exclude the presence of another, hitherto unidentified, genetic variant in our pedigree which could be responsible for the clinical phenotype. Another limitation is the absence of additional genetic testing of the affected family (e.g., the parents of the mutation-carrying father). However, our in vitro electrophysiology findings suggest a deleterious effect of E280K on Kv4.3 functioning (altered activation, inactivation, time to peak, and deactivation properties). Further supported by the known pathogenicity of other KCDN3 variants described in the literature, we would consider that our functional data strongly suggest a pathogenic effect of Kv4.3 E280K (evidence corresponding to PS3 functional studies and PP2 variant spectrum according to the guidelines of the American College of Medical Genetics and Genomics and the Association for Molecular Pathology [26]), although the associated phenotype appears to be relatively mild for this particular variant.

## 4. Materials and Methods

### 4.1. Ethical Statement

The adult patient provided consent to this study on his and the children’s behalf. *Xenopus laevis* oocytes were extracted surgically, in accordance with ethical permits granted by the Swedish National Board for Laboratory Animals and the regional ethical committee, Stockholms Norra Djurförsöksetiska nämnd (N686/21).

### 4.2. Clinical Evaluation and Genetic Analysis 

The index case and her twin sister were evaluated by a pediatric neurologist (NG) at the Department of Pediatrics, Rijnstate Hospital, Arnhem, The Netherlands. Information about the father and the brother was gathered from medical charts. Cognitive evaluation was performed using the Wechsler Preschool and Primary Scale of Intelligence [27]. Whole exome sequencing was performed using the Agilent V5 kit (Agilent Technologies, Santa Clara, CA, USA), followed by in silico analysis for 357 genes associated with movement disorders (see Appendix A for screened genes).

### 4.3. Molecular Biology 

cDNA encoding human *KCND3* wildtype (Kv4.3 WT), *KCND3* E280K (Kv4.3 E280K), and KCNIP2 (KChIP2) in pXOOM [28] were acquired from GenScript (Piscataway, NJ, USA). Plasmids were linearized (*KCND3* WT and E280K; XbaI, KCNIP2; XhoI) prior to in vitro transcription using the T7 mMessage mMachine kit (Ambion, Austin, TX, USA). cDNA and cRNA concentrations and purity were determined using a Nanodrop spectrophotometer (ThermoFisher Scientific, Waltham, MA, USA). 

### 4.4. Oocyte Preparation

The surgical procedure for *Xenopus laevis* oocyte extraction as well as RNA injection were conducted in line with previous descriptions [29]. 24 h after extraction, oocytes were injected individually with (a) KChIP2 (70 pg) and (b) either Kv4.3 WT (70 pg; “WT”), Kv4.3 WT + Kv4.3 E280K (35 + 35 pg; “WT/E280K”), or Kv4.3 E280K (70 pg; “E280K”). Oocytes were incubated at 12 °C for six days prior to recordings. 

### 4.5. Parallel Oocyte Voltage-Clamp Recordings

Electrophysiology experiments were conducted using the parallel eight-channel, two-electrode OpusXpress 6000A voltage clamp (Molecular Devices, San José, CA, USA) [30]. To minimize series resistances in the voltage clamp circuit, micropipettes with tip resistances of 0.3–1 MOhm for the current-conducting electrode and 0.3–2 MOhm for the voltage-recording electrode were used. Recordings were performed at 22 °C in a 1 mM [K+] solution (consisting of 88 mM of NaCl, 1 mM of KCl, 0.8 mM of MgCl_2_, 0.4 mM of CaCl_2_, and 15 mM of HEPES, adjusted to pH 7.4 with NaOH). Continuous perfusion, mediated by Minipuls 3 peristaltic pumps (Gilson, Madison, WI, USA), was maintained at 1.5 mL/min. Data were sampled at 10.4 kHz using OpusXpress 1.10.42 (Molecular Devices) software. 

### 4.6. Data Analysis 

For the electrophysiological data, baseline currents (at holding potentials) were subtracted from the peak currents and the time of peak was identified using Clampfit 10 software (Molecular Devices). One-way and two-way ANOVA were used to compare findings. Correction for multiple testing was conducted using the Holm–Sidak method. Alpha < 0.05 was considered significant. Statistical analysis and rendering were conducted in GraphPad Prism 6.0 (GraphPad Software, San Diego, CA, USA). Data are presented as means ± SEM. The Boltzmann sigmoidal equation (Equation (1) below) was used for fitting baseline-subtracted and peak-normalized conductance data to quantify activation and inactivation.
*Y* = 1/(1 + exp((*V*_1/2_ − *X*)/*Slope*))(1)
where *Y* is the conductance amplitude as a fraction of 1, *X* is the membrane potential in mV, *V*_1/2_ is the membrane potential of half-maximal conductance, and *Slope* is the steepness of the curve.

## Figures and Tables

**Figure 1 ijms-24-10924-f001:**
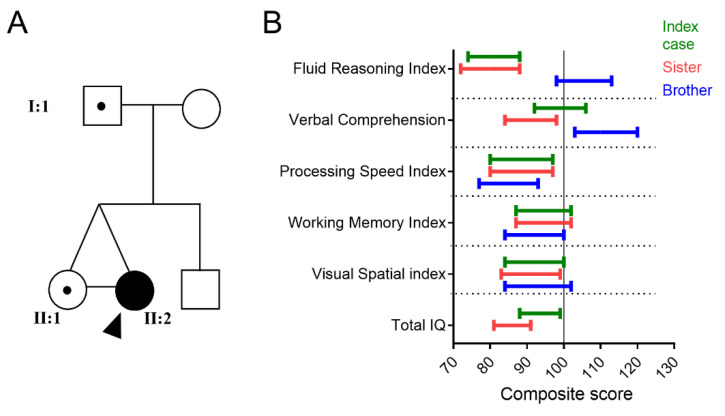
Pedigree and cognitive performance of the three children. (**A**) The father and two monozygotic twins are carriers of the E280K variant. Arrowhead points to index case. The brother and the mother are not carriers. (**B**) Result intervals in cognitive tests of the index patient (green), the twin sister (red), and the brother (blue).

**Figure 2 ijms-24-10924-f002:**
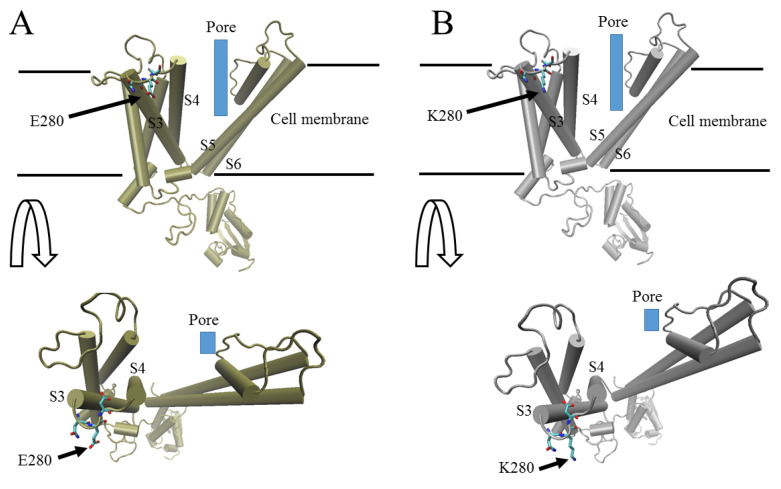
Location of E280 in the Kv4.3 protein. The panels show one subunit, whereas four subunits form the functional channel. (**A**) Kv4.3 WT with E280 and (**B**) Kv4.3 E280K mutation, showing residue 280 in the second extracellular linker as colored sticks. The location of the ion-conducting pore between S5 and S6 is indicated in blue. For clarity, the panels show one single Kv4.3 subunit, whereas the complete functional channel is made up of four subunits.

**Figure 3 ijms-24-10924-f003:**
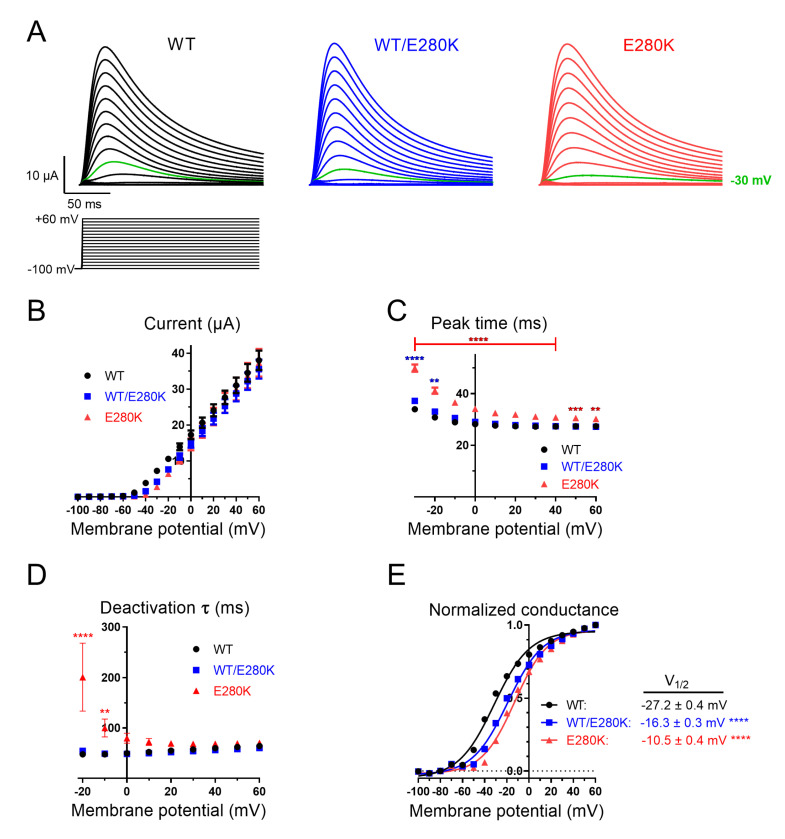
Kv4.3 WT/E280K/KChIP2 current–voltage and conductance–voltage relationships. (**A**) Representative currents for Kv4.3 WT (black), WT/E280K (blue), and E280K (red), elicited by voltage steps from −100 to +60 mV. The holding potential was −100 mV. Voltage steps lasted 1 s and were applied in 10 mV increments. The interpulse interval was 5 s. Note the reduced response at −30 mV for WT/E280K and E280K, compared to WT (green traces). (**B**) I–V relationships for (**A**), measured at the peak current. (**C**) Time to peak current following voltages that induce measurable activation under all three conditions. (**D**) Deactivation time constant for the inactivating current, derived from an exponential decay function fitted over 150 ms. Asterisks in (**C**,**D**) indicate significant differences relative to WT. **, *p* < 0.01; ***, *p* < 0.001; and ****, *p* < 0.0001, two-way ANOVA with Holm–Sidak correction for multiple comparisons. (**E**) Normalized g–V relationships for (**A**), measured at the peak current. Half-activation voltages of normalized conductance, derived from fits of the Boltzmann function to data are shown in panel E. WT; *n* = 17 (Boltzmann slope factor = 16.5 ± 0.3), WT/E280K; *n* = 12 (Boltzmann slope factor = 16.4 ± 0.3), E280K; *n* = 10 (Boltzmann slope factor = 15.5 ± 0.3). Asterisks indicate significant differences relative to WT. ****, *p* < 0.0001, one-way ANOVA with Holm–Sidak correction for multiple comparisons.

**Figure 4 ijms-24-10924-f004:**
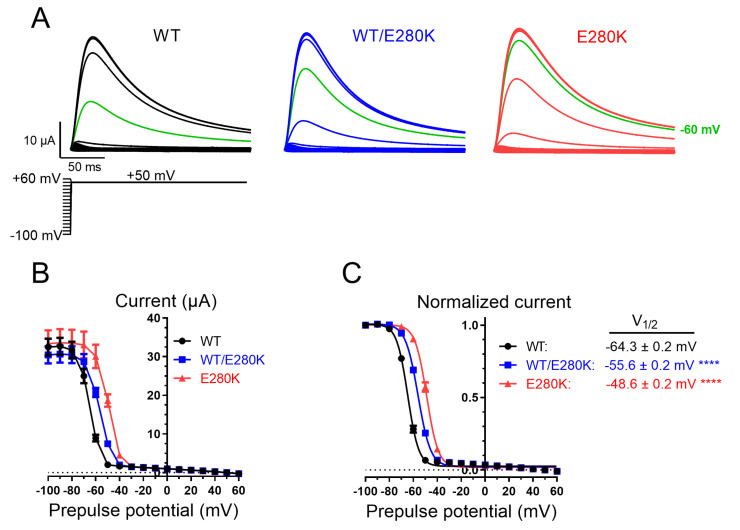
Kv4.3 WT/E280K/KChIP2 steady-state inactivation relationships. (**A**) Representative current traces of Kv4.3 WT (black), WT/E280K (blue), and E280K (red) at +50 mV. The voltage protocol includes 1 s prepulse steps (same as shown in Figure 1). Note the greater current amplitude following prepulse potentials of −60 mV for WT/E280K and E280K, compared to WT (green traces). (**B**) Steady-state inactivation currents for (**A**), measured at the peak current. (**C**) Normalized steady-state inactivation relationships for (**A**), measured at the peak current. Half-maximal normalized steady-state inactivation values derived from fitting a Boltzmann sigmoidal function to the data in (**C**) are shown. The holding potential was −100 mV. Each prepulse step was +10 mV and lasted 1 s. The interpulse interval was 5 s. WT; *n* = 17 (Boltzmann slope factor = −4.8 ± 0.1), WT/E280K; *n* = 12 (Boltzmann slope factor = −5.4 ± 0.1), E280K; *n* = 10 (Boltzmann slope factor = −5.1 ± 0.2). Asterisks indicate significant differences relative to WT. ****, *p* < 0.0001, one-way ANOVA with Holm–Sidak correction for multiple comparisons.

## Data Availability

Anonymized data will be shared by the investigators upon request.

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
