# Peer review of "An E280K Missense Variant in KCND3/Kv4.3—Case Report and Functional Characterization"

_ijms, 2023, doi:10.3390/ijms241310924_

Round 1
Reviewer 1 Report
In this manuscript, the authors reported and characterize a newly discovered mutation located on the S3-S4 extracellular loop of the Kir4.3 channel. This mutation was identified in a five-year-old girl experiencing episodes of headaches, clumsiness, and disruptions in her gait. Intriguingly, the identical mutation was also detected in her asymptomatic father and twin sister. This case study makes a valuable contribution to our current comprehension of spinocerebellar ataxia's pathogenesis, a condition previously associated with different Kir4.3 channel mutations. The manuscript is well-structured and logically organized.
Comments:
1. Figures 3 & 4: The current amplitudes, reaching as high as tens of μA, raise concerns about the efficacy of the membrane voltage clamp. Have potential serial resistance errors been taken into consideration?
2. Figures 3 & 4: The fitting definitions for voltage-dependent activation and inactivation need refining. For G/V relationships, the Boltzmann function, a type of sigmoidal function, should be utilized. The function's equation and coefficients should be clearly defined and displayed. In addition to V1/2, the Hill slope should also be presented in the results.
3. Besides voltage-dependent activation and steady-state inactivation, the analysis of a novel disease-linked mutation should also include deactivation.
4. Figure 3C: Is there a significant difference between the Time to peak current of E280K and WT at negative membrane potentials? Statistical analyses would enhance the clarity of this point.
5. Figures 3E & 4D: Please annotate the statistical analysis results directly on Figures 3E and 4D for better readability.
6. Lines 134-138: V1/2 values of the steady-state inactivation-voltage relationship should initially be presented as mean ± SEM, prior to stating the changes observed.
7. Please clarify which ANOVA method was employed for statistical analysis.
8. In mammalian cells, WT Kv4.3 channels exhibit rapidly inactivating A-type potassium currents, whereas the inactivation kinetics in oocytes are significantly slower. Given that E280K is a mutation linked to human disease, it is crucial to study the biophysical properties of this mutation in mammalian cell models using patch clamping. If patch clamping studies are unfeasible within the lab or through collaboration, it would be beneficial to include a discussion addressing this limitation.
Author Response
Reviewer 1
In this manuscript, the authors reported and characterize a newly discovered mutation located on the S3-S4 extracellular loop of the Kir4.3 channel. This mutation was identified in a five-year-old girl experiencing episodes of headaches, clumsiness, and disruptions in her gait. Intriguingly, the identical mutation was also detected in her asymptomatic father and twin sister. This case study makes a valuable contribution to our current comprehension of spinocerebellar ataxia's pathogenesis, a condition previously associated with different Kir4.3 channel mutations. The manuscript is well-structured and logically organized.
We appreciate the feedback provided by Reviewer 1. Please find our responses detailed below (blue).
Comments:
- Figures 3 & 4: The current amplitudes, reaching as high as tens of μA, raise concerns about the efficacy of the membrane voltage clamp. Have potential serial resistance errors been taken into consideration?
For Xenopus oocyte voltage-clamp, we used 3M KCl micropipettes with tip resistances 0.3-1 MOhm for the current-conducting electrode and 0.3-2 MOhm for the voltage-recording electrode. The high current amplitudes are consistent for all conditions and likely reflect a high channel expression level. By using high expression levels to study large currents, we were able to better investigate the small differences in conductance and inactivation. While we cannot rule out the possibility of voltage errors due to series resistance, any such errors would equally affect WT and E280K mutant channels due to the similar current amplitudes. We have now commented on this in the Methods and Discussion sections.
- Figures 3 & 4: The fitting definitions for voltage-dependent activation and inactivation need refining. For G/V relationships, the Boltzmann function, a type of sigmoidal function, should be utilized. The function's equation and coefficients should be clearly defined and displayed. In addition to V1/2, the Hill slope should also be presented in the results.
We have now fitted the Boltzmann function (as is now described in the Methods) to the G/V data and report the V1/2 in the figures. We now also report the slope factors in the figure legends.
- Besides voltage-dependent activation and steady-state inactivation, the analysis of a novel disease-linked mutation should also include deactivation.
We thank the reviewer for this comment and have now added data on the deactivation phase for the three conditions. Similar to the time of peak, deactivation (measured as the time constant for an exponential decay function) is slower for E280K compared to WT, in the 0-30 mV interval.
- Figure 3C: Is there a significant difference between the Time to peak current of E280K and WT at negative membrane potentials? Statistical analyses would enhance the clarity of this point.
We thank Reviewer 1 for the valuable comment. We have performed statistical analyses and identified a significantly longer time to peak for E280K compared to WT, at -30 mV (p = 0.0036, two-way ANOVA). This has been added to the Results.
- Figures 3E & 4D: Please annotate the statistical analysis results directly on Figures 3E and 4D for better readability.
Statistical analyses are now presented in the figures.
- Lines 134-138: V1/2 values of the steady-state inactivation-voltage relationship should initially be presented as mean ± SEM, prior to stating the changes observed.
The steady-state inactivation V1/2 values are now presented as means ± SEM before the changes are commented upon.
- Please clarify which ANOVA method was employed for statistical analysis.
We have detailed the use of one-way ANOVA for statistical analyses of activation and inactivation V1/2. Adjustment for multiple testing was performed using the Holm-Sidak method. Two-way ANOVA was used for peak time and deactivation time constant, and this is now also described.
- In mammalian cells, WT Kv4.3 channels exhibit rapidly inactivating A-type potassium currents, whereas the inactivation kinetics in oocytes are significantly slower. Given that E280K is a mutation linked to human disease, it is crucial to study the biophysical properties of this mutation in mammalian cell models using patch clamping. If patch clamping studies are unfeasible within the lab or through collaboration, it would be beneficial to include a discussion addressing this limitation.
We agree that patch-clamp electrophysiology using mammalian cell lines would add value considering the swifter inactivation kinetics. This is unfortunately outside the scope of the current investigation/collaboration. Instead, as suggested, we comment on this limitation in the Discussion.
Reviewer 2 Report
The Authors report a five-year-old girl with headache attacks, clumsiness, and a history of transient gait disturbances. A NGS analysis covering 357 genes associated with movement disorders identified a KCND3 novel variant (c.838G>A,p.E280K) which is present in the proband, in her identical twin-sister and in the father. KCND3 pathogenic variants have been associated with spinocerebellar ataxia type 19/22 (SCA19/22) and cardiac arrhythmias. The proband and her sister show only mild developmental delay with impaired fluid reasoning, but not ataxia or parkinsonism. The Authors also performed electrophysiology experiments in Xenopus oocyte expressing the E280K, and the data indicate that Kv4.3 E280K affects channel activation and inactivation, whereas peak current amplitudes were similar for WT/WT, WT/E280K, and E280K/E280K, suggesting that the E280K may be considered to be relatively benign.
The report of a new KCND3 variant, associated with a milder clinical presentation, may be interesting, but I have major remarks on the manuscript in the present form.
Page1, line 44. The bibliography, is not up-to-date. At least 22 variant have been reported and not a dozen, as cited. The bibliography need to be updated.
Page 2 line 60, page 6 line 202. It is not clear how the genetic analysis was performed: the Authors state: “Exome sequencing for 357 genes….”. It was an Whole Exome Sequencing and than an insilico analysis for the 357 genes performed? Or a NGS-targeted resequencing analysis was performed using a probe-based customized gene panel including the 357 genes? With which kit?
In general, authors should be more cautious in considering the KCND3 E280K variant the cause of the mild developmental delay of the two sisters. The variant is present in the father and in the 2 sisters; the father and the proband showed episodic gait disturbances, which is not present in the other sister. On the other hand, the 2 sisters present mild developmental delay, which is not present in the father. It is therefore not clear, with which phenotype the variant is really correlating. The functional studies show only a mild deficit in channel functioning. The Author should discuss the possibility of another genetic cause in the family and on which basis they consider pathogenic the variant, according to the ACMG classification (Richards et al, 2015). The discussion should better assess the problem of variable penetrance in KCND3-related disorders.
Moreover, it would be interesting to test for the presence of the variant the parents of the father to assess if the variant arose denovo in the father.
Author Response
Reviewer 2
The Authors report a five-year-old girl with headache attacks, clumsiness, and a history of transient gait disturbances. A NGS analysis covering 357 genes associated with movement disorders identified a KCND3 novel variant (c.838G>A,p.E280K) which is present in the proband, in her identical twin-sister and in the father. KCND3 pathogenic variants have been associated with spinocerebellar ataxia type 19/22 (SCA19/22) and cardiac arrhythmias. The proband and her sister show only mild developmental delay with impaired fluid reasoning, but not ataxia or parkinsonism. The Authors also performed electrophysiology experiments in Xenopus oocyte expressing the E280K, and the data indicate that Kv4.3 E280K affects channel activation and inactivation, whereas peak current amplitudes were similar for WT/WT, WT/E280K, and E280K/E280K, suggesting that the E280K may be considered to be relatively benign.
The report of a new KCND3 variant, associated with a milder clinical presentation, may be interesting, but I have major remarks on the manuscript in the present form.
We are grateful for the feedback provided by Reviewer 2. Please find our detailed responses below (blue).
Page1, line 44. The bibliography, is not up-to-date. At least 22 variant have been reported and not a dozen, as cited. The bibliography need to be updated.
We thank the reviewer for this comment and have now updated the bibliography to include the most recently presented ataxia-associated KCND3 variants.
Page 2 line 60, page 6 line 202. It is not clear how the genetic analysis was performed: the Authors state: “Exome sequencing for 357 genes….”. It was an Whole Exome Sequencing and than an insilico analysis for the 357 genes performed? Or a NGS-targeted resequencing analysis was performed using a probe-based customized gene panel including the 357 genes? With which kit?
The genetic analysis was performed using Whole Exome Sequencing followed by in silico analysis. The Agilent V5 kit was used.
In general, authors should be more cautious in considering the KCND3 E280K variant the cause of the mild developmental delay of the two sisters. The variant is present in the father and in the 2 sisters; the father and the proband showed episodic gait disturbances, which is not present in the other sister. On the other hand, the 2 sisters present mild developmental delay, which is not present in the father. It is therefore not clear, with which phenotype the variant is really correlating. The functional studies show only a mild deficit in channel functioning. The Author should discuss the possibility of another genetic cause in the family and on which basis they consider pathogenic the variant, according to the ACMG classification (Richards et al, 2015). The discussion should better assess the problem of variable penetrance in KCND3-related disorders.
Moreover, it would be interesting to test for the presence of the variant the parents of the father to assess if the variant arose denovo in the father.
We agree that the index patient shows a mild phenotype compared to previously reported KCND3 variants associated with ataxia. This may align with our electrophysiology findings of E280K on Kv4.3 channels, which suggest altered function, rather than a complete loss of function. We also cannot exclude that other variant genes may be responsible for the atypical phenotype. This is now added to the discussion. Also, further discussion regarding the variable penetrance is added. Nevertheless, we consider that KCND3 is a gene in which missense variation is a common cause of disease, giving supporting evidence for pathogenicity (PP2, variant spectrum according to ACMG). Furthermore, we would argue that the functional effects that we observe constitute strong evidence for pathogenicity (PS3, functional studies according to ACMG). We agree that it would have been interesting and relevant to evaluate additional family members, although this is outside the scope of the current study.